# Prognostic Profiling of the EMT-Associated and Immunity-Related LncRNAs in Lung Squamous Cell Carcinomas

**DOI:** 10.3390/cells11182881

**Published:** 2022-09-15

**Authors:** Qifeng Sun, Yan Gao, Yehui Zhang, Hongmei Cao, Jiajia Liu, Shi-Yong Neo, Keguang Chen, Yanping Bi, Jing Wu

**Affiliations:** 1Department of Thoracic Surgery, Shandong Provincial Hospital Affiliated to Shandong First Medical University, Jinan 250021, China; 2Department of Pharmacy, the Second Hospital of Shandong University, Jinan 250033, China; 3Department of Clinical Pharmacy, The First Affiliated Hospital of Shandong First Medical University & Shandong Provincial Qianfoshan Hospital, Shandong Engineering and Technology Research Center for Pediatric Drug Development, Shandong Medicine and Health Key Laboratory of Clinical Pharmacy, Jinan 250014, China; 4Department of Pharmacy, Zhangqiu District People’s Hospital, Jinan 250200, China; 5School of Pharmaceutical Sciences, Shandong University, Jinan 250012, China; 6Singapore Immunology Network, Agency for Science, Technology and Research, Singapore 138668, Singapore

**Keywords:** lncRNA, EMT, subtypes, prognosis, lung squamous cell carcinomas

## Abstract

Lung squamous cell carcinoma (Lung SCC) is associated with metastatic disease, resulting in poor clinical prognosis and a low survival rate. The aberrant epithelial–mesenchymal transition (EMT) and long non-coding RNA (LncRNA) are critical attributors to tumor metastasis and invasiveness in Lung SCC. The present study divided lncRNAs into two subtypes, C1 and C2 (Cluster 1 and Cluster 2), according to the correlation of EMT activity within the public TCGA and GEO databases. Subsequently, the differential clinical characteristics, mutations, molecular pathways and immune cell deconvolution between C1 and C2 were evaluated. Lastly, we further identified three key lncRNAs (DNM3OS, MAGI2-AS3 and LINC01094) that were associated with EMT and, at the same time, prognostic for the clinical outcomes of Lung SCC patients. Our study may provide a new paradigm of metastasis-associated biomarkers for predicting the prognosis of Lung SCC.

## 1. Introduction

Lung squamous cell carcinoma (Lung SCC) is a major subtype of NSCLC, accounting for approximately 40% of all lung cancer patients [1]. Even though the diagnosis and therapies have made tremendous progress over the past decades, the average 5-year survival rate is <5% due to high rates of recurrence and metastasis [2]. Alongside the development of a variety of targeted therapies over the years, the prognosis of Lung SCC patients could be vastly improved [3]. However, in reality, only a small proportion of Lung SCC are sensitive to such targeted drugs for actual beneficial objective response [4,5]. Therefore, there is an unmet need to identify the essential biomarkers and specific targets underlying the progression of Lung SCC to develop more effective remedies.

The development and progression of Lung SCC are driven by several complex factors, including morphological, molecular and genetic alterations [6,7]. Among them, the epithelial–mesenchymal transition (EMT) represents a typical driver of cellular plasticity during polyclonal cancer progression [8,9]. Tumor cells derived from epithelial cells represent both mesenchymal and epithelial characteristics, which are necessary for the invasive and metastatic properties of carcinoma cells [10]. High occurrences of metastasis are usually associated with EMT in most malignant epithelial tumors [11]. Hence, EMT could be a promising biomarker for the prognostic evaluation of Lung SCC [12].

Long noncoding RNAs (LncRNAs) are a group of transcripts including more than 200 nucleotides that regulate various genes’ expression, which are abnormally expressed in several types of cancer. LncRNAs possess a considerable potential in driving tumor progression for its heavy involvement in tumor metastasis and invasiveness [13]. Notably, lncRNAs are described as an important regulatory RNA molecule to influence the EMT process at multiple levels [14]. Nonetheless, the relationship between EMT-associated LncRNAs and its association to Lung SCC tumor progression remained unclear. Using a systematic approach, we investigated a link between lncRNAs and EMT in Lung SCC and illustrate the related signaling transduction pathways and clinical significance.

## 2. Materials and Methods

### 2.1. Access and Pre-Processing of Publicly Available Expression Profile Data

From the TCGA GDC API, the Lung SCC cohort for tumor bulk transcriptomics and clinicopathological information were accessed for downstream analysis. Samples without survival information were removed. Ensembl gene IDs were matched to their respective gene symbol. In addition, the expression data and clinical information were obtained from GSE29013, GSE30219, GSE37745 and GSE50081 datasets in the GEO database (https://www.ncbi.nlm.nih.gov/geo/, accessed on 1 September 2022). We pre-processed the RNA-seq data in GEO as follows: (1) Download standardized data sets; (2) retain the samples with survival time and survival state; (3) retain lung squamous cell carcinoma samples; (4) combine the four data sets and remove Batch Effect (Appendix A); (5) reannotate the GPL570 gene and convert the probe to gene symbol. The clinical data after screening are shown in Appendix A. The annotation file platform of four datasets is GPL570. The EMT-related genes are derived from the EMT pathway (“HALLMARK_EPITHELIAL_MESENCHYMAL_TRANSITION”) in the MSigDB database.

### 2.2. Identification of EMT-Associated LncRNA

For the acquisition of LncRNA expression profiles, the V32 version of the GTF file from the GEN CODE website (https://www.gencodegenes.org/, accessed on 1 September 2022) was used. The TCGA expression spectrum and GEO data are divided into mRNA and lncRNA according to the annotation in the file. Next, we performed a single-sample gene set enrichment analysis (ssGSEA) using the “GSVA” R package, to calculate the EMT score of each sample in TCGA and CGGA datasets, respectively. The gene set “HALLMARK_EPITHELIAL_MESENCHYMAL_ TRANSITION” was downloaded from the MSigDB database for GSVA analysis. Then, Pearson correlation coefficients and *p* values between the EMT score and lncRNA in the corresponding dataset of LUSC samples in the TCGA and GEO databases were calculated, respectively. The resulting correlation analysis between the EMT score and lncRNA are shown in Appendix A: EMT, LncRNA, CSV, based on the threshold filtering of |cor| > 0.3 and *p* < 0.05.

### 2.3. Recognition *of EMT-Associated LncRNA Subtypes*

The samples were classified by clustering [15] using a consistency matrix constructed by Consensus Cluster Plus. The EMT-associated lncRNA subtypes of the samples were obtained by using the screened lncRNA associated with the EMT score. In this study, we used the KM algorithm and Euclidean as a measure of distance to perform 500 bootstraps, and each bootstrap process contained 80% of the patients in the training set. In addition, we set the clustering number as 2 to 10, which was used to determine the optimal classification by calculating the consistency matrix and consistency cumulative distribution function.

### 2.4. Gene Set Enrichment Analysis (GSEA) and Functional Annotation

For the discovery of pathways related to different molecular subtypes and biological processes, we used “GSEA” to analyze the pathways. We performed gene set enrichment analysis (GSEA) using all candidate gene sets in the Hallmark database [16]. ClusterProfiler packages were applied for feature annotations.

### 2.5. Analysis of Transcription Factor Activity

The score of TF activity was performed in accordance with the developed method by Garcia-Alonso [17]. Analysis of variance (ANOVA) was used to compare TF activation levels among the clusters. *p* < 0.05 was set as the selected conditions for significant difference transcription factors.

### 2.6. First Order Partial Correlation Analysis

A first-order partial correlation was performed to explore interlinks among lncRNAs, glycolysis scores and glycolysis-associated genes. The glycolysis score was assumed to be x, and glycolysis-associated gene expression was y. The first-order partial correlation between x and y conditioned on lncRNAs was:$$r_{xylncRNA}=\frac{r_{xy}-r_{xlncRNA}*r_{ylncRNA}}{\sqrt{\left(1-r_{xlncRNA}^{2}\right)*\left(1-r_{ylncRNA}^{2}\right)}}$$

### 2.7. Risk Model Establishment

We calculated the risk score of each one using the formula: Score = (betai × Expi), i represents the expression level of EMT-associated lncRNA, and beta is the gene coefficient in univariate Cox regression of corresponding lncRNA. Patients were divided into high- and low-risk groups according to the optimal segmentation points. Survival curves were drawn by the Kaplan–Meier method for prognostic analysis, and the significant differences was determined by the logarithmic rank test.

### 2.8. Real Time Qualitative-PCR

Total RNAs were extracted from fresh tumor tissues using a total RNA Kit (Cat. No. R6834, Omega bio-tek, Norcross, GA, USA) and reversely transcribed using a HiFiScript cDNA Synthesis Kit (Cat. No. CW2569, CWBIO, Taizhou, China). Reverse transcription was performed at 42 °C for 15 min, followed by 85 °C for 5 min to inactivate the enzyme activity. Samples were subjected to qPCR using a Thermofisher QuantStudio Real-Time PCR System (Thermofisher, Shanghai, China): 10 µL reaction containing 2× UltraSYBR Green Mixture (Cat. No. CW0957S, CWBIO, Taizhou, China), 50 nM forward and reverse primers and 0.5 µL cDNA. The qPCR protocol was executed for 40 cycles and each cycle consisted of denaturation at 95 °C for 15 s, annealing and extension at 60 °C for 1 min. The sequences of the primers were as follows: human β-actin F: ’CATGTACGTTGCTATCCAGGC’; R: ’CTCCTTAATGTCACGCACGAT’. DNM3OS F: ’TTCATTGCCAGTTCCCCAGTC’; R: ’ACTGAGACACACTCAAGGGC’. MAGI2-AS3 F: ’ATAACAGAATGCAGGAGAGCACA’; R: ’TGCTGTCCCTGGCTCTTGAA’. LINC01094 F: ’TTGTTTGGCAGGCACTCCAT’; R: ’TGTTGTCTCACCACCAGCAG’.

### 2.9. Western Blot

The fresh tissues were homogenized in a RIPA buffer containing 1 mM PMSF. Equivalent amounts of protein (30 μg) were subjected to Western blot analysis. Proteins were transferred to polyvinylidene difluoride membranes (Millipore, Billerica, MA, USA) and immunoblotted with the indicated antibodies as follows: E-Cadherin antibody (Cat. No. A20798, ABclonal, Wuhan, China), N-Cadherin antibody (Cat. No. A19083, ABclonal, China) and β-actin antibody (Cat. No. K200058M, Solarbio, Beijing, China). After being incubated overnight at 4 °C, the membranes were washed and incubated with the secondary antibody (Cat. No. ZB-2301, ZSGB-Bio, China and Cat. No. ZB-2305, ZSGB-Bio, Beijing, China). The washed membranes were visualized using a chemiluminescence reagent (Millipore, Billerica, MA, USA) and exposed to X-ray film. Data were quantitated using the Image J Software.

### 2.10. Ethics Statement 

The study was performed in accordance with the provisions of the Declaration of Helsinki for research involving human subjects and was approved by the Ethical Review Committee of the second hospital of Shandong University (approval number: KYLL-2021(KJ)P-0401). Written informed consent was obtained from all participants after the fullest explanation of the purpose and procedures of the study.

### 2.11. Statistical Analysis

Statistical computations were performed using GraphPad Prism 8.0 (GraphPad, San Diego, CA, USA). Statistical analyses of results except transcriptome analysis were performed using the standard two-tailed Student’s *t*-test. Data are presented as mean ± SEM. *p* < 0.05 was considered statistically significant, *p* < 0.01 was very significant, *p* < 0.001 and *p* < 0.0001 were extremely significant.

## 3. Results

### 3.1. Systematic Establishment of LncRNA Molecular Typing Based on EMT Scores

The Cancer Genome Atlas (TCGA) and Gene Expression Omnibus (GEO) databases were used to analyze the relationship between EMT scores and lncRNA expression. As shown in Figure 1A, 608 lncRNAs in TCGA and 51 lncRNAs in GEO were identified as significantly changed according to EMT signature. The relatively low overlap indicated that EMT-associated lncRNAs have poor consistency in different platforms. Next, we chose 31 EMT-associated lncRNAs in these two queues for further analysis. ConsensusClusterPlus was adopted to cluster the Lung SCC samples in the TCGA database, and the optimal number of clusters was confirmed according to the cumulative distribution function (CDF). The CDF Delta area curve showed that Cluster 2 presented a relatively stable clustering (Figure 1B). Ultimately, we selected k = 2 to obtain the targeted two molecular subtypes, named C1 (*n* = 210) and C2 (*n* = 283) (Figure 1C and Appendix A).

Further analysis of the prognostic features of these two subtypes demonstrated significant prognostic differences: C2 has a better prognosis while C1 has a worse prognosis (Figure 1D). To substantiate this funding, similar results were validated from the GEO dataset (Figure 1E and Appendix A). On the basis of these data, the two molecular subtypes of lncRNAs related to EMT activity possess portability in diverse cohort studies. In addition, we compared the EMT scores in different molecular subtypes to determine if heterogeneity existed among them. Figure 1F,G demonstrated that the EMT activity of these two subtypes had significant variations both in TCGA and GEO: EMT scores were the highest in the C1 subtype while the lowest in the C2 subtype. Collectively, we demonstrated that the EMT-associated molecular subtyping was feasible and could be established systematically.

### 3.2. Comparison of EMT-Associated LncRNAs Subtypes and Clinical Information

Factors such as age, gender, tumor-stage, node-stage, metastasis-stage and tobacco history always could generally influence the clinical outcomes of LUSC patients. To address if such factors were different comparing the two subtypes of EMT-associated lncRNAs, we compared the distribution of these clinical characteristics in the C1 versus C2 group from the TCGA cohort. These profiling data indicated that no significant differences exist in such factors: including age, tumor-stage, node-stage, metastasis-stage and stages (Figure 2A–E). Of note, the gender (Figure 2F) and tobacco (Figure 2G) groups displayed significant differences comparing the C1 and C2 subtypes.

### 3.3. Mutation Characteristics of the EMT-Associated LncRNAs Subtypes

Next, we investigated which of the EMT-associated lncRNAs subtypes are associated with other tumor genomic alterations in the TCGA-Lung SCC cohort. High levels of aneuploidy score, homologous recombination defects, fraction altered, number of segments and tumor mutation burden are presented in both the C1 and C2 subtypes (Figure 3A). Furthermore, we performed the correlation analysis between EMT activity and the indicators including aneuploidy score, homologous recombination defects, fraction altered, number of segments and non-silent mutation rate, revealing that the activity of EMT is negatively correlated with homologous recombination defects and fraction altered (Figure 3B). Likewise, correlations between EMT-associated subtypes and several gene mutations were observed. Among these genes, the proportions of CDH10, NFE2L2, TPTE, GRMS, DCAF4L2, SPATA31D1, OTOF, ZNF521 and LRRTM4 mutations were relatively lower in C1 than C2. Conversely, FAT3, NRXN1, MYHS, PPP1R3A and AKAP9 mutation rate proportions were higher in C1 than C2 (Figure 3C). These data support that C1 was mainly enriched EMT and cancer-associated pathways, while C2 was related to hypoxia and the cell cycle. The details on mutation quantity and significance are shown in Appendix A. Taken together, the results suggest that EMT and cancer-associated pathways could be the likely driver for the poor clinical prognosis seen in the C1 subtype.

### 3.4. Pathway Analysis of EMT-Associated LncRNAs Subtypes

To provide further mechanistic insights on EMT-associated lncRNAs in Lung SCC, we next looked into the differential gene set enrichment of various signaling pathways between C1 and C2. We have utilized all candidate gene sets from the MSigDB Hallmarks Database [16] for gene set enrichment analysis (GSEA). Using a defined threshold of FDR <0.05 for significance, 21 pathways were activated while 11 pathways were inhibited in C2 subtypes compared to C1 subtypes within the TCGA–LUSC cohort. Likewise, in the GEO dataset, 25 pathways were activated while 9 pathways were inhibited in C2 compared to C1. On the whole, the major activated pathways included EPITHELIAL_MESENCHYMAL_TRANSITION, INFLAMMATORY_RESPONSE, TNFA_SIGNALING_VIA_NFKB, HYPOXIA, INTERFERON_ALPHA_RESPONSE and ANGIOGENESI, et al. (Figure 4A).

Additionally, we compared the common upregulated pathways both in the C1 and C2 subtypes from the TCGA cohort (Figure 4B) and the GEO cohort (Figure 4C), respectively. From the GSEA analysis of different subtypes, we found that EMT, Immune regulatory pathways (such as INFLAMMATORY_RESPONSE, INTERFERON_ALPHA

_RESPONSE and INTERFERON_GAMMA_RESPONSE) and the cell cycle were upregulated significantly. Therefore, we speculated that lncRNAs existing in those subtypes may play a key role in linking the EMT signaling, immunosuppressive microenvironment and a critical involvement in the cell cycle.

### 3.5. The Immune Characteristics of EMT-Associated LncRNAs

To uncover potential association of the tumor immune landscape with EMT-associated lncRNAs, various immune gene signatures were used to evaluate the immune cell infiltration in three Lung SCC cohorts. The marker genes in immune cells were derived from the literature [18]. The deconvoluted distribution of immune cells in the different EMT-associated lncRNAs subgroups in both TGCA and GEO cohorts are shown in Figure 5A. The expression of various immune gene signatures in the C1 subtype were higher than those in the C2 subtype, including CD8 T cells, DC, Eosinophils, Mast cells and T cells, etc. In addition, ESTIMATE was also used to access the immune cell infiltration (Figure 5B). The “Immune Score” of the C1 subtype in TGCA and GEO cohorts was greater than in the C2 subtype, indicating that there was a relatively stronger immune cell infiltration in the C1 subtype. Furthermore, when patients were divided into high and low immune infiltration groups based on unsupervised hierarchical clustering, most C1 patients belonged to the high immune infiltrations group, while some of the C2 group were affiliated with the low immune infiltrations group and others with the high immune infiltrations group (Figure 5C). Aligned with previous reports, our results showed that high TIL infiltration was associated with more prominent EMT characteristics [19]. Differences in the immune response existed in tumors according to their EMT status and tumor types, due to a complex tumor environment.

### 3.6. Differential Analysis of Immunotherapy in EMT-Associated LncRNAs Subgroups

In this section, we analyzed whether there were differences in the subgroups of EMT-associated lncRNAs for predicted responses to immunotherapy. We first compared the differential gene expression of various immune checkpoints comparing the two different subtypes of EMT-associated lncRNAs. The immune checkpoints were obtained from the database HisgAtlas [20]. In general, there were significant differences in the expression of most immune checkpoints comparing the two subtypes with a large portion of immune checkpoints highly expressed in the C1 subtype. (Appendix A). In both the TCGA and GEO datasets, some of the highly expressed immune checkpoints included CD244, HAVCR2, CD47, CD80, CTLA4, PDCD1, ICOS and IDO1 (Figure 6A,B). Notably, CTLA4, PDCD1, CD80 and IDO1 were much upregulated in the C1 subtype compared to the C2 subtype, indicating that the immune landscape in C1 could be more immunosuppressive than that in C2. Next, we utilized TIDE (http://tide.dfci.harvard.edu/, accessed on 1 September 2022) software, which is a prediction model for response to immunotherapy. Within the TCGA cohort, a higher TIDE score was observed in the C1 subtype as compared to C2, which indicates a higher possibility of immune escape in C1 and less benefit from immunotherapy (Figure 6C). Likewise, similar results in the TIDE score and immunotherapy response were shown in the GEO cohort (Figure 6D).

### 3.7. Characteristic Analysis of EMT-Associated LncRNAs

Next, we investigated the relationship between tumor molecular characteristics and the EMT-associated lncRNAs. We observed that most EMT-associated lncRNAs were negatively correlated with EMT activity compared to protein-coding genes (Figure 7A). The application of the lnc-ATLAS database further revealed that most lncRNAs associated with EMT activity were sub-cellularly located in the nucleus (Figure 7B). Among them, we determined that 62.64% of RCIs were negative (RCI < 0) in the TCGA cohort and 71.43% of RCIs were negative (RCI < 0) in the GEO cohort. To explore whether there were differences in transcription factor (TF) activity between the two defined EMT-associated lncRNAs subgroups, we calculated TF activity scores in TCGA and GEO, respectively, according to the method developed by Garcia-Alonso [17]. Variance analysis (ANOVA) was used to compare the TF activation levels between clusters to identify TFs with significant differences. The results showed that there were 139 and 115 significantly different TFs in TCGA and GEO cohorts, respectively. The differential analysis results of TFs were shown in Appendix A. We subsequently investigated the association between EMT-associated lncRNAs and dysregulated TFs. Figure 7C shows that a set of TFs was negatively correlated with EMT-associated lncRNAs gene expression. We identified 31 lncRNAs correlated with EMT, among which 14 and 17 were positively or negatively correlated with EMT, respectively. From a selection of key lncRNAs, we further explored the differences in TFs activity from molecular isoforms (Figure 7D) and found that 15 TFs upregulated significantly in the C1 subtype, displayed in Appendix A. The distribution of upregulated TFs activities in the C1 subtype from the TCGA database is shown in Figure 7E. The similar distribution of TFs activities from the GEO database is displayed in Appendix A. From there, we speculated that the uniformly upregulated TFs might be associated with a poorer prognosis in C1 and further performed a functional enrichment analysis of TFs. The results showed that TFs targeted genes were significantly enriched to some cancer pathways, including the PI3K-Akt signaling pathway, Epstein-Barr virus infection, the JAK-STAT signaling pathway, and the HIF-1 signaling pathway, as shown in Appendix A. The activation of these pathways was related to tumor progression and poor prognosis, implying, on the other hand, the EMT-associated lncRNA might play a key role in the activation of cell cycle pathways and some cancer pathways (Figure 7F,G).

### 3.8. Identification of Key EMT-Associated LncRNAs

The potential involvement of lncRNAs in the co-expression of EMT-related genes was analyzed and observed to change significantly even after the data were adjusted to account for the influence of the associated lncRNAs. Based on the EMT score, EMT-related genes and lncRNA gene expression levels, three key lncRNAs were identified according to first-order partial correlation analysis, namely, DNM3OS, MAGI2-AS3 and LINC01094 (Figure 8A). Moreover, the correlation between EMT score and EMT-related genes decreased significantly, indicating that these three identified lncRNAs play an important role in the signaling pathways of EMT and EMT-related genes. Next, we identified the common EMT-related genes that were significantly enriched in the AGE-RAGE signaling pathway in diabetic complications, focal adhesion and ECM−receptor interaction (Figure 8B). More importantly, we were able to use these three key lncRNAs to stratify patients into high- and low-risk groups, which further highlights the prognostic values of the expression of these three lncRNAs (Figure 8C,D). Finally, we used qPCR to verify the expression of the three key lncRNAs in human tissues and normal tissues. The data showed higher expression of lncRNA DNM3OS, MAGI2-AS3 and LINC01094 in LUSC tumor samples compared with normal tissues (Figure 9A). In contrast, the lower expression of E-cadherin, which is a classical phenomenon of EMT, was confirmed in tumor tissues at the protein level via Western blot (Figure 9B,C).

Despite the development of various novel therapeutic strategies, Lung SCC maintained a high fatality rate due to the poor prognosis [21], which could be attributed to multiple elements, including metastasis, immune infiltration and cell cycle alterations [22,23]. EMT is a transformation of epithelial cells into mesenchymal cells, which is a critical biological process for the migration of cells in the embryo [24]. In cancer biology, EMT is known to contribute to cellular plasticity and for enhancing tumor metastatic potential [25]. LncRNAs are considered to be associated with various tumors, serving as biomarkers to predict cancer prognosis, including lung cancer. However, it remained elusive whether the expression of EMT-associated lncRNAs could be prognostic in the evaluation of tumor progression in Lung SCC.

The present study screened for the EMT-associated lncRNAs within the TCGA and GEO patient datasets and classified them into C1 and C2 subtypes. We found that there were significant prognostic differences between the two subtypes in both the TCGA and GEO cohorts. Along with a good prognosis in C2 and a poor prognosis in C1, C2 had the lowest EMT score while C1 had the highest EMT score. We then focused on the comparison of different characteristics in these two molecular subtypes. In TCGA cohorts, gender and tobacco were attributes that were significantly different when comparing C1 and C2 subtypes. Subsequent genomics analysis also showed that homologous recombination defects and fraction altered were significantly correlated with the two subtypes and EMT activities.

In general, the EMT and cancer-associated pathways mainly enriched in C1 resulted in a poor prognosis, while proliferation and the cell cycle are enriched in the C2 subtype in Lung SSC. Our mutational analysis revealed that FAT3, a FAT atypical cadherin family protein [26], had a higher mutation rate in C1. FAT3 mutation is related to tumor mutation burden and poor prognosis in several cancers [27], which could also be related to the finding that the C1 subtype is associated with tumor metastatic potential and invasiveness. The CDH10 mutation was reported to be significantly associated with better overall survival, independent of tumor–node–metastasis staging [28]. In our study, the mutation rate of CDH10 was much higher in C2 than in C1, indicating a better prognosis in C2.

Further analysis revealed that signaling pathways including EMT, interferon response and the G2M-checkpoint were all upregulated in C1. Interestingly, we found that several immune gene scores were much higher in C1 than C2. In general, high immune infiltrations often predict for better treatment outcome [29]. On the other hand, the complex tumor microenvironment could be characterized with EMT that is also coupled with high immune infiltrations resulting in a poorer prognosis [30], which is similarly observed in our findings as exhibited by the C1 subtype. Typical immune checkpoints. including HAVCR2 (Tim-3) [31], CD80 [32], ICOS [33], CD47 [34], PDCD1 [35], were much higher in C1 than C2, indicating a higher likelihood of immunosuppression in the C1 subtype.

Our present study identified 31 common lncRNAs associated with EMT activity in both lung squamous cancer sets that were also recognized in EMT-signaling transmission in various cancers. The functions of lncRNAs are closely related to their subcellular localization [36]. In general, lncRNAs in the nucleus activate or repress the expression of target genes by binding to them directly and also participate in the regulation of gene expression by engaging in histone modification or recruitment of transcription factors [37]. Nevertheless, lncRNAs in the cytoplasm can also interact with miRNAs as a competitive endogenous RNA to participate in the regulation of target gene expression. Most lncRNAs associated with EMT activity were located in the nucleus [38], demonstrating that EMT-associated lncRNAs would regulate specific TFs in Lung SCC. LncRNAs, which are located in the cell nucleus, interacted directly with TFs to promote gene expression to play important roles in chromatin organization, transcription, and regulation of posttranscriptional gene expression [39,40]. Therefore, we focused on the correlation analysis of lncRNAs located in the nucleus with differential TFs. We observed a TFs collection negatively correlated with EMT-associated lncRNAs and identified a group of TFs upregulated in C1. Functional enrichment analysis demonstrated that the TFs enrichment signal pathways were associated with the tumor progression and prognosis. From here, we propose that EMT-associated lncRNAs might activate tumor-associated pathways to promote the tumor progression through regulating specific TFs. We further identified three key lncRNAs, namely, DNM3OS, MAGI2-AS3 and LINC01094, that could be used in combination to stratify patients into high- or low-risk groups for the evaluation of prognostic outcomes. To substantiate our findings from the TCGA and GEO dataset analyses, we further verified that the expression of the three key lncRNAs to be upregulated in LUSC tumor samples compared to normal samples and, at the same time, the classic EMT phenotypes, including E-cadherin [41] and N-cadherin, were observed to downregulate and upregulate, respectively. Nonetheless, there are still limitations in the present study. While we extrapolated that certain EMT-related lncRNAs could be potentially valuable biomarkers in Lung SSC based on the TCGA and GEO clinical datasets, future directions could seek to answer more complex research questions, such as the underlying mechanisms behind the regulation of EMT-related lncRNAs and the specific critical downstream targets that trigger EMT and other potential immune-modulating pathways. Further mechanistic studies should at the same time consider therapeutic interventions targeting these lncRNAs to treat cancers.

## 4. Conclusions

In essence, our findings demonstrate that (1) the two subtypes, stratified based on EMT-associated lncRNA in Lung SCC cohorts, had significant differences in Lung SCC prognosis; (2) the two subtypes carried different clinical characteristics, mutation characteristics, pathway characteristics and immune characteristics; (3) three crucial lncRNAs, connected to EMT and tumor necrosis pathways, were identified to predict the prognosis of Lung SCC. Our present study provided novel insight into the biology underlying EMT-associated lncRNA and how these lncRNA could be applied as a potential biomarker for LUSC tumor progression.

## Figures and Tables

**Figure 1 cells-11-02881-f001:**
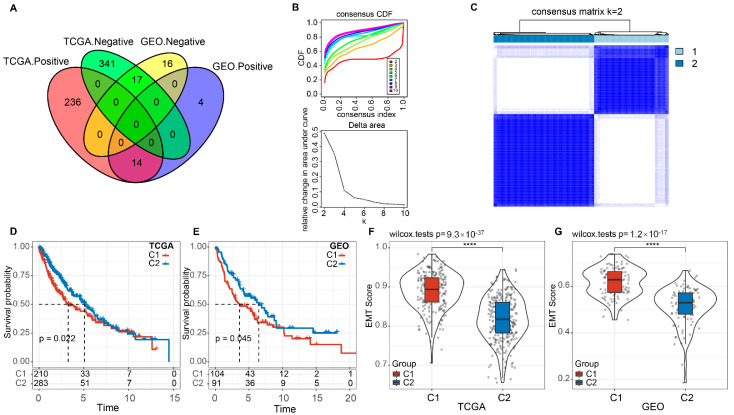
The EMT-associated lncRNAs subtypes in TCGA. (**A**) The intersection Venn diagram between EMT activity-related lncRNAs in the TCGA and GEO arrays. (**B**) The CDF curve and CDF Delta area curve in TCGA queue samples. (**C**) The clustering heat map of TCGA samples with Consensus K = 2. (**D**) The OS-time prognostic survival curve of EMT-associated lncRNAs subtypes in TCGA. (**E**) The OS-time prognostic survival curve of molecular subtypes in GEO. (**F**) The differences in EMT activity between the two molecular subtypes in the TCGA cohort. (**G**) The differences in EMT activity between the two molecular subtypes in the GEO cohort. Ns; *p* > 0.05; **** *p* < 0.0001.

**Figure 2 cells-11-02881-f002:**
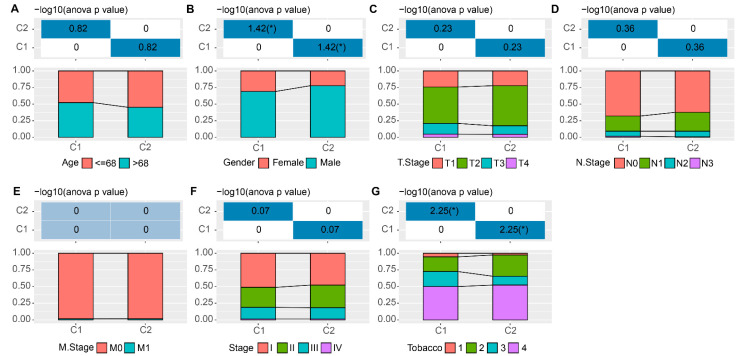
Clinical information distribution of molecular subtypes in the TCGA cohort. The chi-square test of the clinical information distribution of different molecular subtypes: including age (**A**), T-Stage (**B**), N-Stage (**C**), M-Stage (**D**), stage (**E**), gender (**F**) and tobacco (**G**). Tobacco lifelong non-smoker (less than 100 cigarettes smoked in lifetime) = 1; Current smoker (includes daily smokers and non-daily smokers or occasional smokers) = 2; Current reformed smoker for > 15 years (greater than 15 years) = 3; Current reformed smoker for ≤15 years (less than or equal to 15 years) = 4. * 0.01 ≤ *p* < 0.05.

**Figure 3 cells-11-02881-f003:**
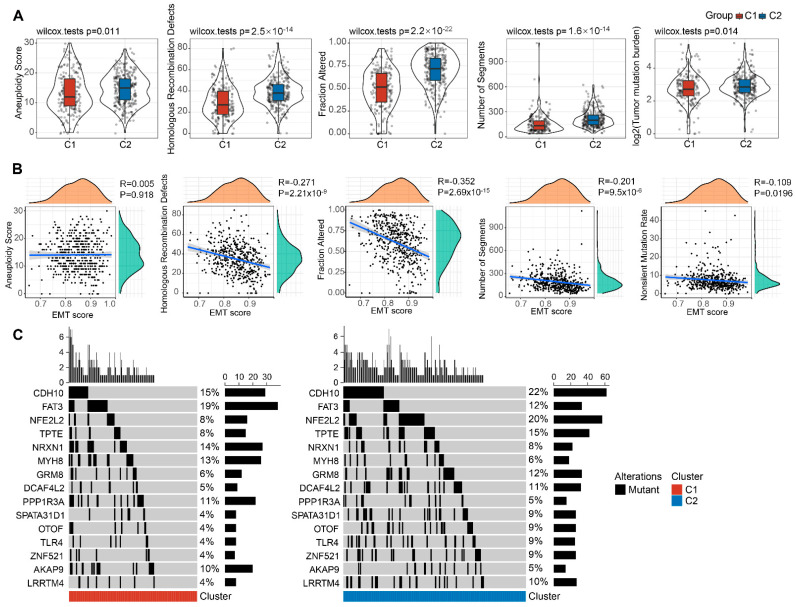
Genomic alterations of the molecular subtypes in TCGA cohorts. (**A**) The discrepancy of homologous recombination defects, aneuploidy score, fraction altered, number of segments and tumor mutation in the TCGA molecular subtypes. (**B**) Correlation analysis between EMT activity and homologous recombination defects, aneuploidy score, fraction altered, number of segments, and tumor mutation in TCGA cohorts. (**C**) The analysis of somatic mutations in the subtypes (Fisher’s test).

**Figure 4 cells-11-02881-f004:**
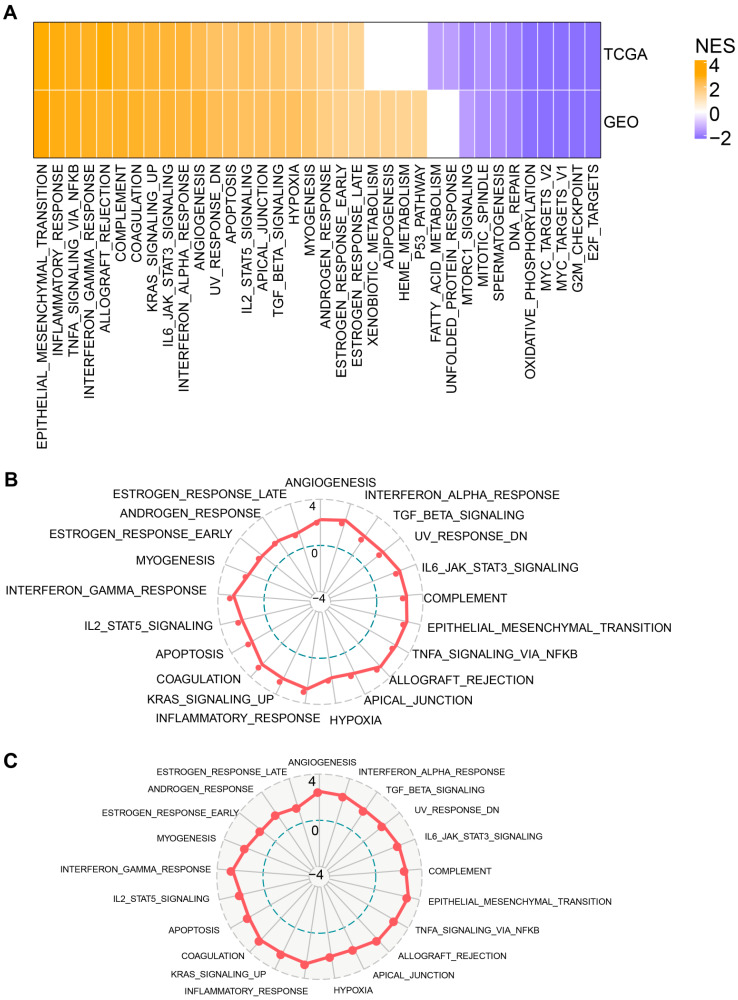
Signaling pathway analysis of the EMT-associated lncRNAs subgroup. (**A**) Heatmap demonstrating normalized enrichment scores (NESs) of Hallmark pathways calculated by comparing cluster 2 with cluster 1 (with a false discovery rate (FDR) of <0.05). Radar plots indicating NESs of Hallmark pathways calculated through a gene set enrichment analysis (GSEA) of cluster 1 versus cluster 2 in the TCGA cohort (**B**) and GEO cohort (**C**).

**Figure 5 cells-11-02881-f005:**
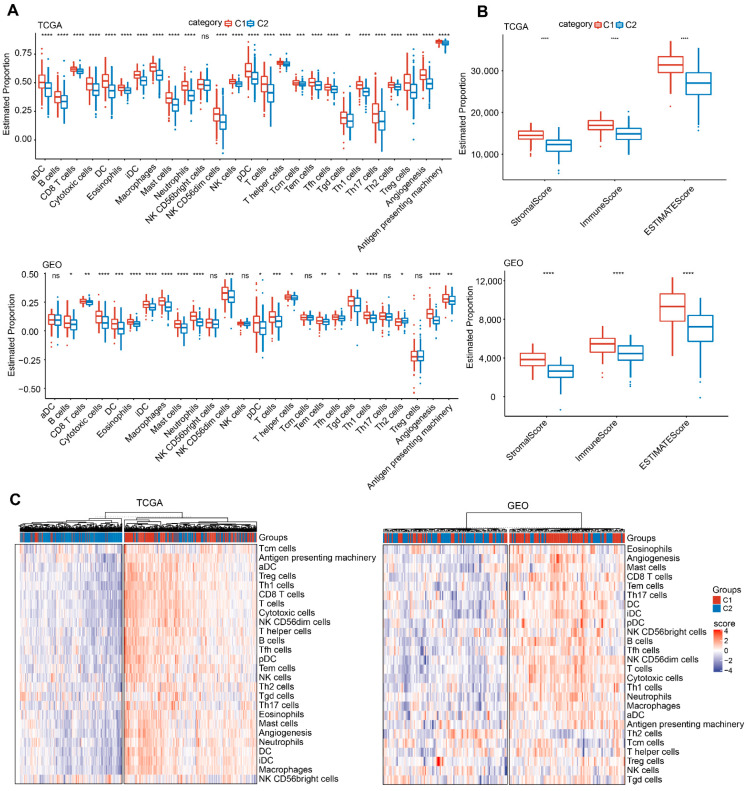
The immune characteristics of EMT-associated LncRNAs. (**A**) Proportion of immune cell components in two Lung SCC cohorts. (**B**) The proportion of immune cell components calculated by ESTIMATE software. (**C**) Unsupervised clustering of distinct immune cell infiltrations in EMT-stratified cluster 1 and cluster 2 cancer patients. Ns; *p*>0.05; * 0.01 ≤ *p* < 0.05; ** *p* < 0.01, *** *p* < 0.001, **** *p* < 0.0001.

**Figure 6 cells-11-02881-f006:**
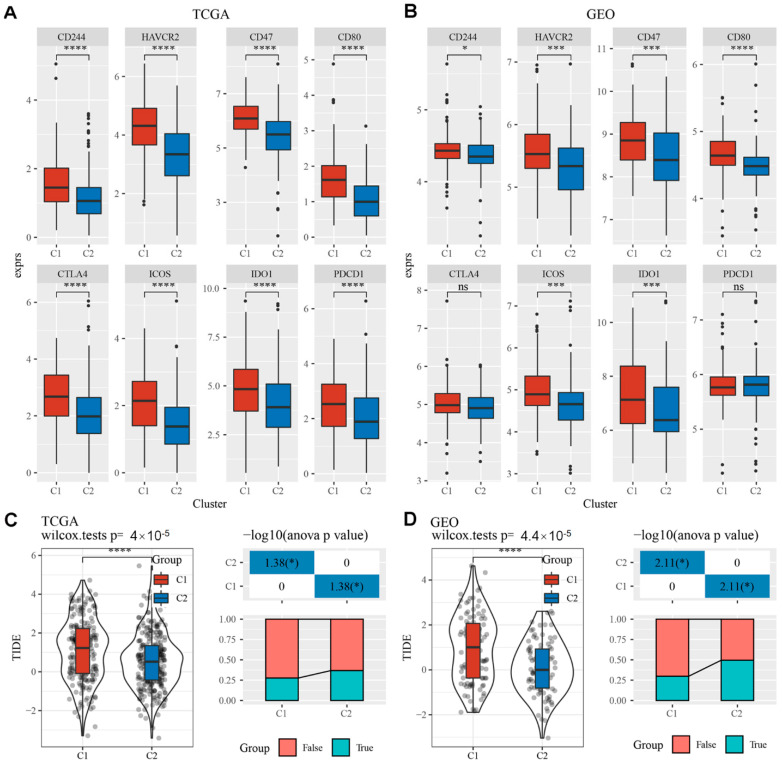
Differential analysis of immunotherapy in EMT-associated lncRNAs subgroups. The boxplots demonstrate immune checkpoints that were upregulated in cluster 1 compared with cluster 2 in TCGA (**A**) and GEO (**B**). (**C**) Differences in TIDE scores and immune response status among different molecular subtypes of TCGA. (**D**) Differences in TIDE score and immune response status among different molecular subtypes in GEO cohort. Ns; *p* > 0.05; * 0.01 ≤ *p* < 0.05; *** *p* < 0.001, **** *p* < 0.0001.

**Figure 7 cells-11-02881-f007:**
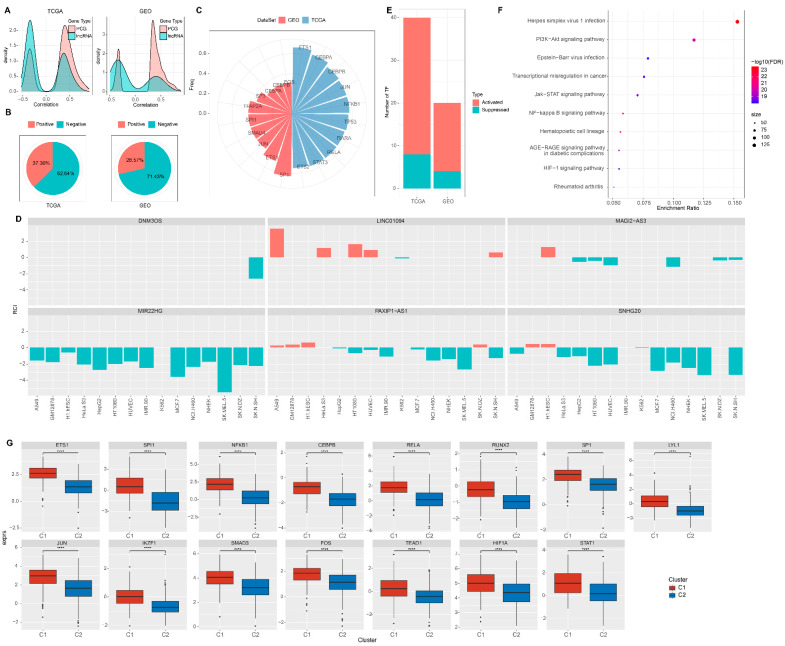
Characteristic analysis of EMT-associated lncRNAs. (**A**) Correlation coefficient density curve of EMT-associated lncRNAs and protein-coding gene (PCG). (**B**) Definition of EMT-associated lncRNAs in cells, where negative was RCI < 0 and positive was RCI > 0. (**C**) Distribution of TF significantly negatively correlated with nuclear lncRNAs in the two cohorts. (**D**) Distribution of lncRNAs in cells consistent with EMT activity in the two LSCC. (**F**) Distribution of activated and inhibited TF in the C1 subtype compared with the C2 subtype. (**G**) Functional enrichment analysis of uniformly upregulated TF in the C1 subtype. (**E**) Distribution of uniformly upregulated TF in TCGA subtypes. Ns; *p* > 0.05; **** *p* < 0.0001.

**Figure 8 cells-11-02881-f008:**
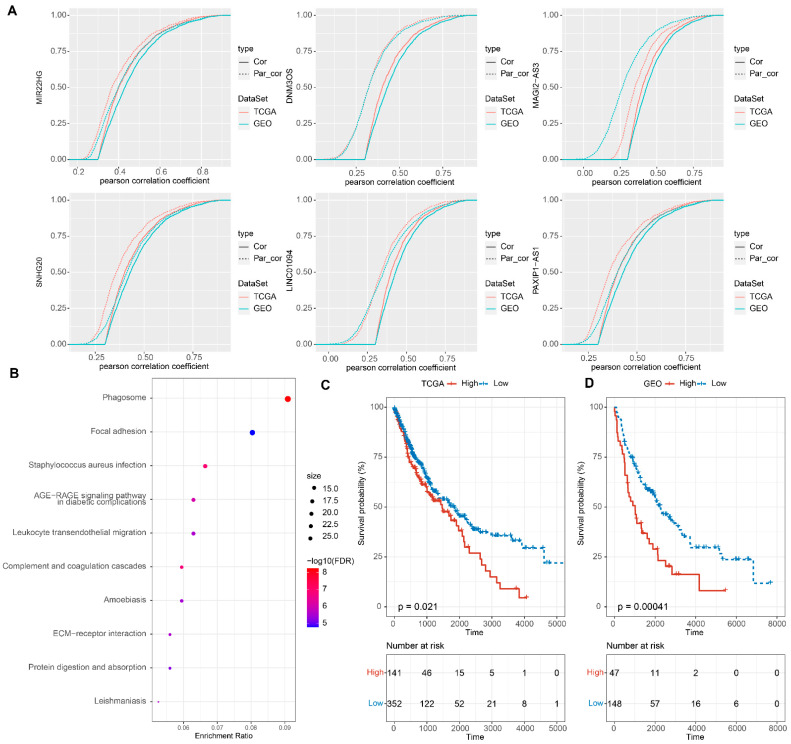
Identification of key EMT-associated lncRNAs. (**A**) CDF with or without adjustment for EMT-associated lncRNAs by using a first order partial correlation. Solid lines indicate CDFs of correlation coefficients between glycolysis scores and gene expressions without adjustment. Dashed lines indicate first-order partial correlation-adjusted relations between glycolysis scores and gene expressions. These two distributions were compared using the Kolmogorov–Smirnov test. The *x*-axis represents Pearson correlation coefficients between glycolysis scores and gene expressions, and the *y*-axis represents cumulative probabilities. (**B**) Enrichment analysis of genes significantly associated with lncRNAs. (**C**) KM curves of high- and low-risk groups in the TCGA cohort. (**D**) KM curves of high- and low-risk groups in the GEO cohort.

**Figure 9 cells-11-02881-f009:**
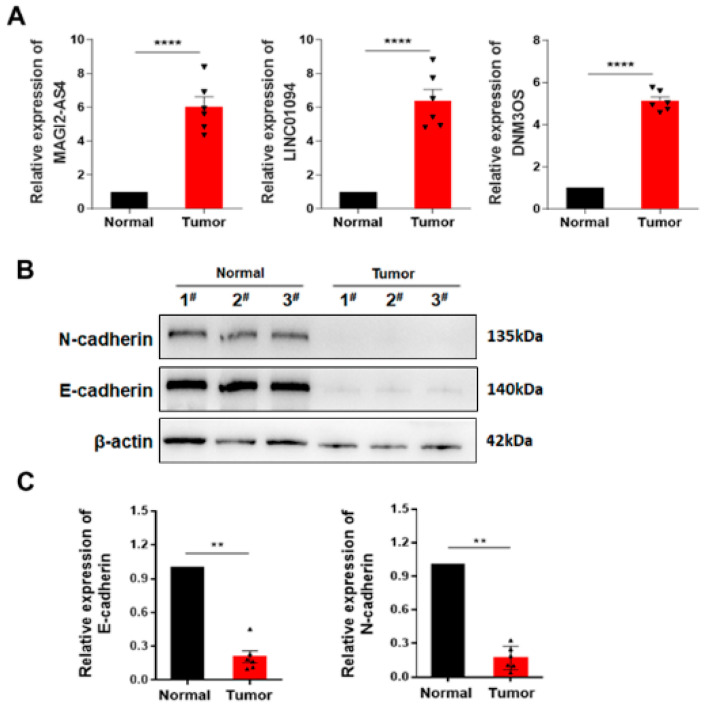
Validation of three EMT-associated lncRNAs in a patient tumor. (**A**) Relative mRNA levels of lncRNA DNM3OS, MAGI2-AS3 and LINC01094 in patient tumor tissues. (**B**,**C**) Western blot analysis of N-cadherin and E-cadherin in human tumor tissues. Ns; *p* > 0.05; ** *p* < 0.01, **** *p* < 0.0001.

## Data Availability

All the data were included in the Appendix A.

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
