# Peer review of "Prognostic Profiling of the EMT-Associated and Immunity-Related LncRNAs in Lung Squamous Cell Carcinomas"

_cells, 2022, doi:10.3390/cells11182881_

Round 1

Reviewer 1 Report

Please check the text in 2.1 “Source of expression profile data”; the sentences do not seem to be complete.

What is the source of the samples? Where can the markers can be found? In blood by blood collection or in tissue by high invasive biopsies? Important factor for biomarkers!

Are the biomarkers specific for LUSC? Can they be found in other cancer types in the body?

3.1: Indicate number of objects per cluster; information can be found in Table S1 but please indicate as number in the text. Is the distribution even between the clusters?

Figure 2: What means 1, 2, 3, and 4 in the tobacco groups?

Figure 3B: the axis labelling is hard to read, also in the original images after enlarging

Please indicate the significance level: normally <0.5 is *; <0.1 is **, and < 0.01 is ***. What does **** (4 asterisks) mean?

Figure 9: the loading control is not optimal; this experiment proves the relevance of the in silico research. How many samples were used – 3 per group? This number should be increased. Moreover, this experiment is not described in the method section.

An observation if these lncRNA also can be found in blood would improve the clinical significance of the biomarkers as the development of the cancer after treatment may be observed and monitored.

Author Response

Authors’ reply to review report (Reviewer 1):

Comments: 1. Please check the text in 2.1 “Source of expression profile data”; the sentences do not seem to be complete.

Response:    We thank the reviewer for these positive comments and accurately understanding of our findings. We also thank the reviewer for carefully reading our manuscript. In the revised manuscript, we have revised it to “Access and pre-processing of publicly available”.

Comments: 2. What is the source of the samples? Where can the markers can be found? In blood by blood collection or in tissue by high invasive biopsies? Important factor for biomarkers! Are the biomarkers specific for LUSC? Can they be found in other cancer types in the body?

Response:    We thank the reviewer for the positive comments. The samples are obtained from the TCGA database, and all samples were highly invasive biopsies (cancer tissues). All of them have done RNA sequencing. We systematically conducted studies based on the EMT-associated and Immunity-related LncRNAs in Lung squamous cell carcinomas in addition to the experimental validations. We found three potential biomarkers for Lung SCC. These biomarkers haven’t been found in other comprehensive studies on other cancer types.

Comments: 3. 3.1: Indicate number of objects per cluster; information can be found in Table S1 but please indicate as number in the text. Is the distribution even between the clusters?

 Response:   We thank the reviewer for these positive comments and accurately understanding of our findings. As suggested, we added the number in the text. C1 accounts for 43% of the samples, whereas C2 accounts for 57%. In terms of the number distributions, we believe the classification was good to cluster the sample into different characteristics.

Comments: 4. Figure 2: What means 1, 2, 3, and 4 in the tobacco groups? 

Response:    We thank the reviewer’s valuable advice. We are sorry for the missing legend. We have complemented the explanations in the figure legend.

Comments: 5. Figure 3B: the axis labelling is hard to read, also in the original images after enlarging

Response:    We thank the reviewer for these positive comments and accurately understanding of our findings. We apologize for the problems, we have enlarged the text of all images to make them much clearer.

Comments: 6. Please indicate the significance level: normally <0.5 is *; <0.1 is **, and < 0.01 is ***. What does **** (4 asterisks) mean?

Response:    We thank the reviewer for reading the manuscript carefully. We are sorry for the missing information. We have added all significance level in all figure legend, **** P < 0.0001.

Comments: 7. Figure 9: the loading control is not optimal; this experiment proves the relevance of the in silico research. How many samples were used – 3 per group? This number should be increased. Moreover, this experiment is not described in the method section.

Response:    We thank the reviewer for these positive comments and accurately understanding of our findings. We also thank the reviewer for carefully reading our manuscript. We used -3 per group in original manuscript and increased another 3 samples in Figure S4 to support our conclusion in. In addition, we described the experiments in the method section.

Comments: 8. An observation if these lncRNA also can be found in blood would improve the clinical significance of the biomarkers as the development of the cancer after treatment may be observed and monitored.

Response:    We appreciate tremendously for the valuable suggestions. We would like to collect and detect the blood samples in the further study.

Reviewer 2 Report

The hypothesis of the study seems to be sound. However, the text is poorly written, there are many grammatical, semantic and syntax errors, The methods and the results appear to be given in detail, however, the data is presented in a complicated manner, which causes the reader to lose their interest. The discussion should also be improved using more evidence based data.

Author Response

Authors’ reply to review report (Reviewer 2):

Comments: The hypothesis of the study seems to be sound. However, the text is poorly written, there are many grammatical, semantic and syntax errors, The methods and the results appear to be given in detail, however, the data is presented in a complicated manner, which causes the reader to lose their interest. The discussion should also be improved using more evidence based data.

Response:    We thank the reviewer for these positive comments and accurately understanding of our findings. We also thank the reviewer for carefully reading our manuscript. We have checked grammatical, semantic and syntax errors carefully and modified the manuscript by an English-native expert. In addition, we revised the manuscript completely as suggested.

Reviewer 3 Report

In this manuscript, Sun and colleagues defined a prognostic signature of long non-coding RNA (lncRNA) associated with lung squamous cell carcinomas (SCC). The research is based on a bioinformatic analysis of data publicly available through the Cancer Genome Atlas (TCGA) and Gene Expression Omnibus (GEO) databases, to identify a relationship between epithelial-mesenchymal transition (EMT) and lncRNA expression in lung SCC. This work led to the identification of three lncRNAs (DNM3OS, MAGI2-AS3 and LINC01094) associated to EMT and tumour necrosis pathways.

This is a bioinformatical work which suffers from the limitations that are frequently observed in this kind of studies. We should keep in mind that quite obviously, the larger data sets become using modern analysis techniques, the higher the chances of observing spurious correlations.

My main criticism for this work concerns the apparent arbitrariness applied to select the topic of interest, here the association between EMT, lncRNAs and lung SCC. Using the same scientific approach, you might investigate the implications between ANY set of genes, ANY cancer, and ANY cellular mechanism. Frankly, if something is discovered, it may be difficult to make sense of it.

The number of lncRNAs found to be associated with EMT (608 from TCGA and 51 from GEO), not surprisingly, is high but it is unclear to me how the authors “choose 31 EMT-associated lncRNAs in these two queues for the further analysis” (page 3). Also, it is not clear the rationale and the procedure the authors used to cluster the lncRNAs in the two groups C1 and C2. This is an important point, because most of the results presented in this manuscript is based on such clustering.

It is well known that these gene expression data are heavily confounded by a variation of cell type composition, local inflammatory events, and many individual features. Therefore, I would only be convinced by this part of results if the authors were able to show that cell mechanisms comprising similar numbers of genes (maybe another type of cancer?) would not provide similar outcomes. The demonstration of similar results in independent cohorts, and in patient’s samples is not verifying anything, as these tumour tissue samples may be similarly confounded, not allowing for causal conclusions. In addition to this last point, it is unclear how the authors validated the three lncRNAs identified, as no information is provided on the origin of the patient samples and controls and the procedure used to isolate the RNA and perform RT-qPCR. More importantly, the ethical statements that normally accompany this kind of study are missing.

Overall, the work, along with the identification of potential prognostic biomarkers, explores various characteristics of the samples from a bioinformatics perspective, that may be helpful for further studies. However, the reading is at times difficult, as if many analyses had been done to augment the work but without a well-defined thread. Clearer writing (both in terms of organization and in terms of explaining the various objectives) would help the reading.I think the conclusions drawn by the authors are far from conclusive, so I would not recommend this manuscript for publication on Cells. 

Minor points:

The correct abbreviation of “squamous cell lung cancer” should be “lung SCC”

The details provided in the “Material and Methods” are very scarce, from which it is not possible to subject all experimental details and results to necessary scrutiny. For example, on page 2, the authors wrote: “2.3.1. TCGA Data preprocessing. We preprocess the RNA-seq data in TCGA as following: Firstly, remove samples without survival information, then match ENSG to GeneSymbol.” The meaning of this procedure is unclear. I still think it should be possible that the present data are meaningful, but I think it is impossible to give a fair judgement at this stage. I would strongly recommend demonstrating the specificity of important statements.

The resolution of figures is too low to make them readable.

Check the format of the bibliography in the main text, to make sure it follows the Cells format. For example, many references are provided as PMID identifiers [PMID: 20427518].

Author Response

Authors’ reply to review report (Reviewer 3):

Comments: 1. The number of lncRNAs found to be associated with EMT (608 from TCGA and 51 from GEO), not surprisingly, is high but it is unclear to me how the authors “choose 31 EMT-associated lncRNAs in these two queues for the further analysis” (page 3). Also, it is not clear the rationale and the procedure the authors used to cluster the lncRNAs in the two groups C1 and C2. This is an important point, because most of the results presented in this manuscript is based on such clustering.

Response:    We thank the reviewer for these positive comments and accurately understanding of our findings. We are also sorry for our unclear explanation. Since the consistency between TCGA and GEO is small, we selected 17 EMT-associated lnRNAs between TCGA and GEO negative correlated, and 14 EMT-associated lnRNAs between TCGA and GEO positive correlated. In total, 31 EMT-associated lncRNAs.

Comments: 2. It is well known that these gene expression data are heavily confounded by a variation of cell type composition, local inflammatory events, and many individual features. Therefore, I would only be convinced by this part of results if the authors were able to show that cell mechanisms comprising similar numbers of genes (maybe another type of cancer?) would not provide similar outcomes.

Response:    We thank the reviewer for these positive comments and accurately understanding of our findings. We systematically conducted studies based on the EMT-associated and Immunity-related LncRNAs in Lung squamous cell carcinomas in addition to the experimental validations. We found three potential biomarkers for lung SCC. These biomarkers haven’t been found in other comprehensive studies on other cancer types.

Although bioinformatics approaches for analyzing bulk RNA sequencing data from the TCGA database have been well established, we admit this study shares some limitations with previous bioinformatics studies. As you mentioned, we didn’t show that the biomarkers are specific to Lung SCC, not other cancer types. The efficacy of these biomarkers will be testified in future cohorts, and pan-cancer studies will applied to validate the specificity of these biomarkers.

Comments: 3. The demonstration of similar results in independent cohorts, and in patient’s samples is not verifying anything, as these tumour tissue samples may be similarly confounded, not allowing for causal conclusions.

Response:    We thank the reviewer for these positive comments and accurately understanding of our findings. We also thank the reviewer for carefully reading our manuscript. We have added the numbers of tumor tissue samples to support our conclusion.

Comments: 4. In addition to this last point, it is unclear how the authors validated the three lncRNAs identified, as no information is provided on the origin of the patient samples and controls and the procedure used to isolate the RNA and perform RT-qPCR. 

Response:    We thank the reviewer for these positive comments and accurately understanding of our findings. We also thank the reviewer for carefully reading our manuscript. We have supplemented the details on the RT-qPCR methods.

Comments: 5. More importantly, the ethical statements that normally accompany this kind of study are missing.

Response:    We thank the reviewer for these positive comments and accurately understanding of our findings. We have added the ethical statements in the manuscript.

Comments: 6. Clearer writing (both in terms of organization and in terms of explaining the various objectives) would help the reading.

Response:    We thank the reviewer for the valuable suggestions. We have checked grammatical, semantic and syntax errors carefully and modified the manuscript by an English-native expert. We hope our improvement will help the readers understanding.

Comments: 7. The correct abbreviation of “squamous cell lung cancer” should be “lung SCC”

Response:    We appreciate for your valuable advice. We have corrected the abbreviation of  “squamous cell lung cancer” into “lung SCC”.

Comments: 8. The details provided in the “Material and Methods” are very scarce, from which it is not possible to subject all experimental details and results to necessary scrutiny. For example, on page 2, the authors wrote: “2.3.1. TCGA Data preprocessing. We preprocess the RNA-seq data in TCGA as following: Firstly, remove samples without survival information, then match ENSG to GeneSymbol.” The meaning of this procedure is unclear. I still think it should be possible that the present data are meaningful, but I think it is impossible to give a fair judgement at this stage. I would strongly recommend demonstrating the specificity of important statements.

Response:    We thank the reviewer for the valuable suggestions. We apologize for our unclear expression, we have rephrased the methods part in the revised manuscript.

Comments: 9. The resolution of figures is too low to make them readable.

Response:    We thank the reviewer for carefully reading our manuscript. We have changed the figures to make it clearer.

Comments: 10. Check the format of the bibliography in the main text, to make sure it follows the Cells format. For example, many references are provided as PMID identifiers [PMID: 20427518].

Response:    We thank the reviewer for carefully reading our manuscript. We have checked and revised the bibliography in the main text follows the Cells format.

Reviewer 4 Report

Main Strength:

In this manuscript, the authors have investigated the role of LncRNAs in Lung squamous cell carcinoma (LUSC) especially in EMT and immune-associated pathways. The authors analyzed data from TCGA and GEO databases and established two lncRNA molecular types (C1 and C2) based on EMT scores. They further performed comparisons in various aspects, including clinical information, mutation characteristics, enriched pathway analysis, immune characteristics, immunotherapy response. Additionally, they identified three key EMT-associated lncRNAs, DNM3OS, MAGI2-AS3 and LINC01094 and discovered significant differences in the prognosis of the high-low risk group. Lastly, they verified the expression of these key lncRNA in their patients samples and found higher expression in tumor tissues compared with normal tissues, along with decreased E-cadherin expression. Overall, the authors identified two subtypes and three crucial lncRNAs, indicating their prediction role as novel and effective biomarker in LUSC development. However, there are some questions needed to be answered:

 1)   In immune characteristics of EMT-associated lncRNAs session,  “In this study, most C1 patients belong to the high immune  infiltrations  group,  while  part  of  C2  affiliates  with  low  immune  infiltrations group and others are for high immune infiltrations group. The results was in line with previous reports that high TIL infiltration was associated with more obvious EMT characteristics [15].” Could the authors discuss more about high TIL associated with EMT characteristics?  As mentioned in [15], The immune microenvironment is complex and immune responses can exert either anti- or pro-tumorigenic effects. For example, tumor-infiltrating lymphocytes (TILs) has been shown to inhibit tumor growth and correlate with improved clinical outcomes. Additionally, higher levels of TILs are associated with better disease-free survival (DFS) and overall survival (OS). However, Tumor-associated macrophages (TAMs) and tumor-infiltrating dendritic cells (TIDCs) could promote tumor growth and metastasis, indicating higher levels of TAMs and TIDCs are strongly associated with poor outcomes.

2) Since the authors have the access to patients’ samples, including tumor and adjacent normal tissues. Could the authors performed additional IHC analysis to verify their conclusion? For example, EMT markers (E-cadherin, N-cadherin, Snail+Slug, Vimentin), subtypes of immune cells (T cells, macrophage). Do these genes show lower expression in tumor tissue compared with normal tissues? For western blot, have the authors also evaluated the expression of other EMT markers?

3) Since three key lncRNA were overexpressed in all patients tumor samples, have the authors performed the follow-up studies for these patients? Response to treatment, relapse or not, survival, and etc. Does these data supported the authors’ conclusion?  

Author Response

Authors’ reply to review report (Reviewer 4):
Comments:    1.   In immune characteristics of EMT-associated lncRNAs session,  “In this study, most C1 patients belong to the high immune  infiltrations  group,  while part  of  C2  affiliates  with  low  immune  infiltrations group and others are for high immune infiltrations group. The results was in line with previous reports that high TIL infiltration was associated with more obvious EMT characteristics [15].” Could the authors discuss more about high TIL associated with EMT characteristics?  As mentioned in [15], The immune microenvironment is complex and immune responses can exert either anti- or pro-tumorigenic effects. For example, tumor-infiltrating lymphocytes (TILs) has been shown to inhibit tumor growth and correlate with improved clinical outcomes. Additionally, higher levels of TILs are associated with better disease-free survival (DFS) and overall survival (OS). However, Tumor-associated macrophages (TAMs) and tumor-infiltrating dendritic cells (TIDCs) could promote tumor growth and metastasis, indicating higher levels of TAMs and TIDCs are strongly associated with poor outcomes.
Response:    We thank the reviewer for these positive comments and accurately understanding of our findings. Differences in the immune response existed in tumors according to EMT status and tumor types, due to complex tumor environment. However, the functional mechanism on the relationship of high TIL and EMT characteristics need further study. We are willing to do this research in our further study. We also appreciate for your valuable suggestions.

Comments:    2. Since the authors have the access to patients’ samples, including tumor and adjacent normal tissues. Could the authors performed additional IHC analysis to verify their conclusion? For example, EMT markers (E-cadherin, N-cadherin, Snail+Slug, Vimentin), subtypes of immune cells (T cells, macrophage). Do these genes show lower expression in tumor tissue compared with normal tissues? For western blot, have the authors also evaluated the expression of other EMT markers?
Response:    We thank the reviewer for these positive comments and accurately understanding of our findings. We also thank the reviewer for carefully reading our manuscript. We have detected the N-cadherin, an important EMT marker, using western blot method, and displayed the results in Figure 9. 
Unfortunately, we could not perform the IHC analysis for loss of paraffin block now. However, we will collect samples and detect the indicator in the further study. Thanks very much for the reviewer’s valuable advice.

Comments:    3. Since three key lncRNA were overexpressed in all patients tumor samples, have the authors performed the follow-up studies for these patients? Response to treatment, relapse or not, survival, and etc. Does these data supported the authors’ conclusion?  
Response:    We appreciate tremendously for the reviewer’s precious suggestion. The results have been verified in 5 cohorts (TCGA and 4 GEO datasets). As suggested, the true predictive effect of the Tfh-based gene signature needs to be estimated in a prospective study. As part of the limitation of our study, we will incorporate an external cohort to testify to the predictive efficacy of these biomarkers in lung SCC in future study.

Round 2

Reviewer 2 Report

Results and methods have sgnificantly been improved. However, the findings are still poorly discussed.

Author Response

Comments: Results and methods have significantly been improved. However, the findings are still poorly discussed.

Response:    We thank the reviewer for these positive comments and. We have added some limitations in the Discussion part. We tremendously appreciate for the reviewer’s valuable suggestions to help us improve our study.

Reviewer 3 Report

The authors addressed most of my concerns, I therefore recommend this manuscript for publication on Cells

Author Response

Comments: The authors addressed most of my concerns, I therefore recommend this manuscript for publication on Cells

Response:    We tremendously appreciate for the reviewer’s recommendation and valuable suggestions.